# Electric control of optically-induced magnetization dynamics in a van der Waals ferromagnetic semiconductor

Freddie Hendriks [1], Rafael R. Rojas-Lopez[1,2], Bert Koopmans [3] & Marcos H. D. Guimarães [1] ✉

Electric control of magnetization dynamics in two-dimensional (2D) magnetic materials is an essential step for the development of novel spintronic nanodevices. Electrostatic gating has been shown to greatly affect the static magnetic properties of some van der Waals magnets, but the control over their magnetization dynamics is still largely unexplored. Here we show that the optically-induced magnetization dynamics in the van der Waals ferromagnet $Cr_2Ge_2Te_6$ can be effectively controlled by electrostatic gates, with a one order of magnitude change in the precession amplitude and over 10% change in the internal effective field. In contrast to the purely thermally-induced mechanisms previously reported for 2D magnets, we find that coherent opto-magnetic phenomena play a major role in the excitation of magnetization dynamics in $Cr_2Ge_2Te_6$. Our work sets the first steps towards electric control over the magnetization dynamics in 2D ferromagnetic semiconductors, demonstrating their potential for applications in ultrafast opto-magnonic devices.

Ever since the experimental confirmation of magnetism in two-dimensional (2D) van der Waals (vdW) materials[1,2], researchers have tried to understand their fundamentals and to utilize their unique properties for new technologies, such as novel spintronic devices for information storage and processing[3–6]. The use of magnetization dynamics is particularly interesting since it provides an energy efficient route to transfer and process information[7–11]. A key challenge in this field, named magnonics, is the effective control over the magnetization and its dynamics using electrostatic means, allowing for energy efficient, on-chip, reconfigurable magnonic circuit elements[12–14]. For conventional (three-dimensional) systems this control has been shown to be very promising to reduce the energy barriers for writing magnetic bits using spin-orbit torques[15,16]. Nonetheless, the effect is still relatively modest[17–20]. In contrast, 2D magnetic semiconductors provide an ideal platform for electric manipulation of magnetization. Their low density of states and high surface-to-volume ratio allow for an effective control over the magnetic parameters in these systems, such as the

magnetic anisotropy and saturation magnetization[21–25]. Additionally, 2D magnetic semiconductors offer a bridge to another exciting field: the combination of optics and magnetism. These materials have shown to possess strong light-matter interaction and high magneto-optic coefficients which strength can be further tuned by the use of vdW heterostructures[6,26–31]. These properties make 2D magnetic semiconductors ideal for the merger of two emerging fields: magnonics and photonics.

Most works on the electric control of magnetization in vdW magnets have focused on their magnetostatic properties, such as the magnetic anisotropy, saturation magnetization and Curie temperature, in both metallic[32–35] and semiconducting[21–25] materials. In contrast, their magnetization dynamics have only recently started to receive more attention, and have been studied using microwave-driven magnetic resonance[36–43], or time-dependent magneto-optic techniques[44–52]. The latter were used on antiferromagnetic bilayer $CrI_3$ to show that its magnetic resonance frequency can be electrically tuned by tens of GHz[53]. Nonetheless, the electric control over the

[1]Zernike Institute for Advanced Materials, University of Groningen, Groningen, The Netherlands. [2]Departamento de Física, Universidade Federal de Minas Gerais, Belo Horizonte, Brazil. [3]Department of Applied Physics, Eindhoven University of Technology, Eindhoven, The Netherlands. ✉e-mail: m.h.guimaraes@rug.nl

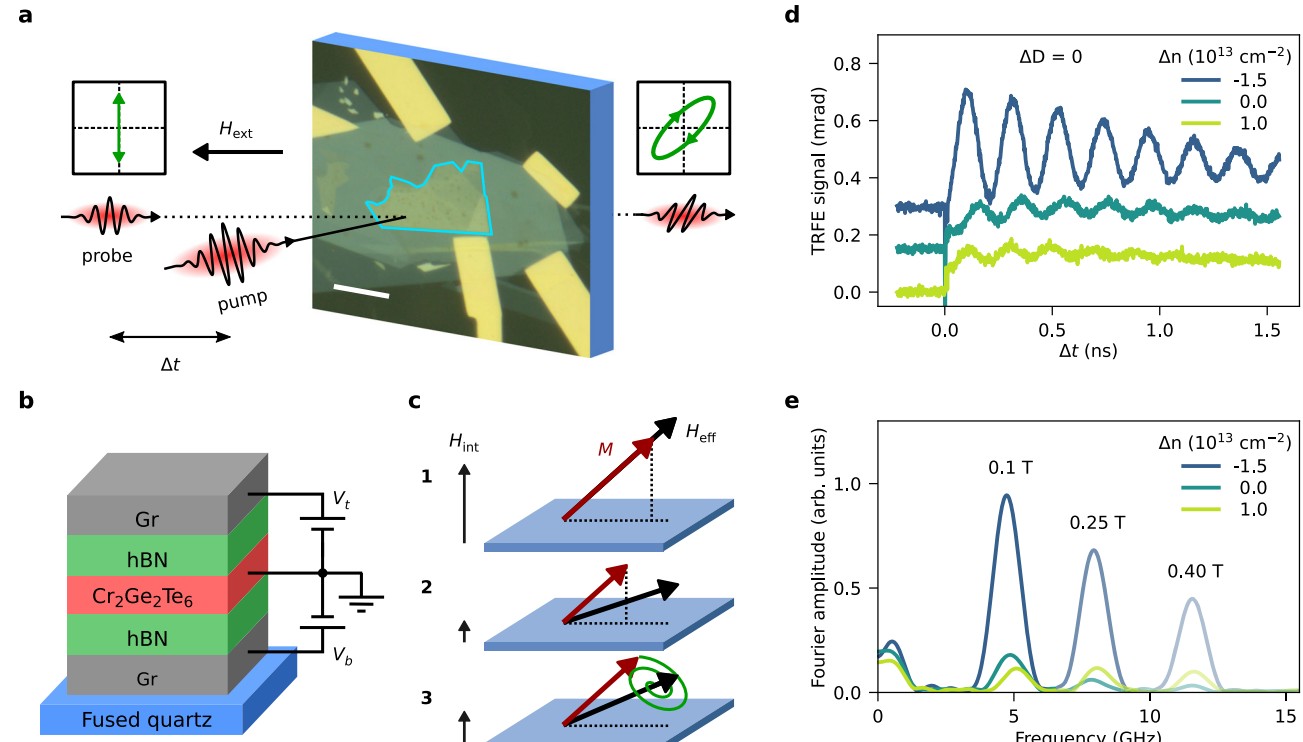

**Fig. 1 | Magnetization dynamics in a CGT based heterostructure. a** Illustration of time-resolved Faraday ellipticity measurements, combined with an optical micrograph of the sample (the scale bar is 10 $\mu m$). The CGT flake is outlined in blue. **b** Schematic of the layers comprising the sample, including electrical connections for gating. **c** Process of laser-induced magnetization precession (see main text). **d** Time-resolved Faraday ellipticity traces (pump-induced change in the Faraday ellipticity) at $\mu_0 H_{ext} = 100$ mT for three different values of $\Delta n$ with $\Delta D = 0$. A vertical offset was added for clarity. **e** Frequency spectrum of the oscillations in the data shown in (**d**). Different transparencies indicate different values of $H_{ext}$.

optical excitation of magnetization and its subsequent dynamics in 2D ferromagnets remains to be explored.

Here we show that the magnetization dynamics of the vdW semiconductor $Cr_2Ge_2Te_6$ (CGT) can be efficiently controlled by electrostatic gating. Using ultrafast (fs) laser pulses we bring the magnetization out of equilibrium and study its dynamics with high temporal resolution through the magneto-optic Faraday effect. Using both top and bottom electrostatic gates, we independently control the gate-induced change in the charge carrier density ($\Delta n$) and the electric displacement field ($\Delta D$) in the CGT, and show that both have drastic effects on the optically-induced oscillation amplitudes and a more modest effect on its frequency. Finally, we observe a strong asymmetric behavior on the magnetization oscillation amplitudes with respect to a reversal of the external magnetic field, which is also strongly affected by both $\Delta n$ and $\Delta D$. This asymmetry can be explained by a strong influence of coherent opto-magnetic phenomena, such as the inverse Cotton-Mouton effect and photo-induced magnetic anisotropy, on the excitation of the magnetization dynamics.

## Results and discussion
### Device structure and measurement techniques
Our sample consists of a 10 nm thick CGT flake, encapsulated in hexagonal boron nitride (hBN), with thin graphite layers as top gate, back gate, and contact electrodes, as depicted in Fig. 1a, b (see Methods for more details). The measurements were performed at low temperatures (10 K), with the sample mounted at 50 degrees with respect to the magnetic field axis for transmission measurements. The laser light is parallel to the magnetic field axis.

We use the time-resolved magneto-optic Faraday effect to monitor the magnetization dynamics in our system using a single-color pump-probe setup similar to the one described in refs. 54,55 (more information in Methods). The process of optical excitation of

magnetization dynamics in van der Waals magnets has been previously reported as purely thermal[44–48,53], similar to many studies on conventional metallic thin films[56–58]. Here we find strong evidence that coherent opto-magnetic phenomena also play an important role in the excitation of the magnetization dynamics. The detailed microscopic description of how the magnetization dynamics is induced is described later in the article, but in short, the excitation of the magnetization dynamics can be described as follows (Fig. 1c): In equilibrium (1), the magnetization **M** points along the total effective magnetic field **H**$_{eff}$, which is the sum of the external field (**H**$_{ext}$), and the internal effective field (**H**$_{int}$) caused by the magnetocrystalline anisotropy ($K_u$) and shape anisotropy. For CGT, **H**$_{int}$ points out-of-plane[2,46,59], meaning that $K_u$ dominates over the shape anisotropy. The linearly polarized pump pulse interacts with the sample (2), reducing the magnetization and changing the magnetocrystalline anisotropy through the mechanisms mentioned above, which causes **M** to cant away from equilibrium. Since **M** and **H**$_{eff}$ are not parallel anymore, this results in a precession of **M** around **H**$_{eff}$, while they both recover to their equilibrium value as the sample cools.

### Gate control of magnetization dynamics
The dual-gate geometry of our device allows for the independent control of both the charge carrier density and the perpendicular electric field. The dependence of $\Delta n$ and $\Delta D$ on the top and back gate voltages—$V_t$ and $V_b$, respectively—is derived in the Methods. The change in the Fermi level induced by $\Delta n$ is expected to affect the magnetic anisotropy of CGT due to the different Cr $d$-orbitals composition of the electronic bands[60]. The effect of $\Delta D$ is, however, more subtle. The inversion symmetry breaking caused by $\Delta D$ can allow for an energy shift of the (initially degenerate) electronic bands, potentially also modulating the magnetization parameters. Additionally, the perpendicular electric field can induce a non-uniform distribution of

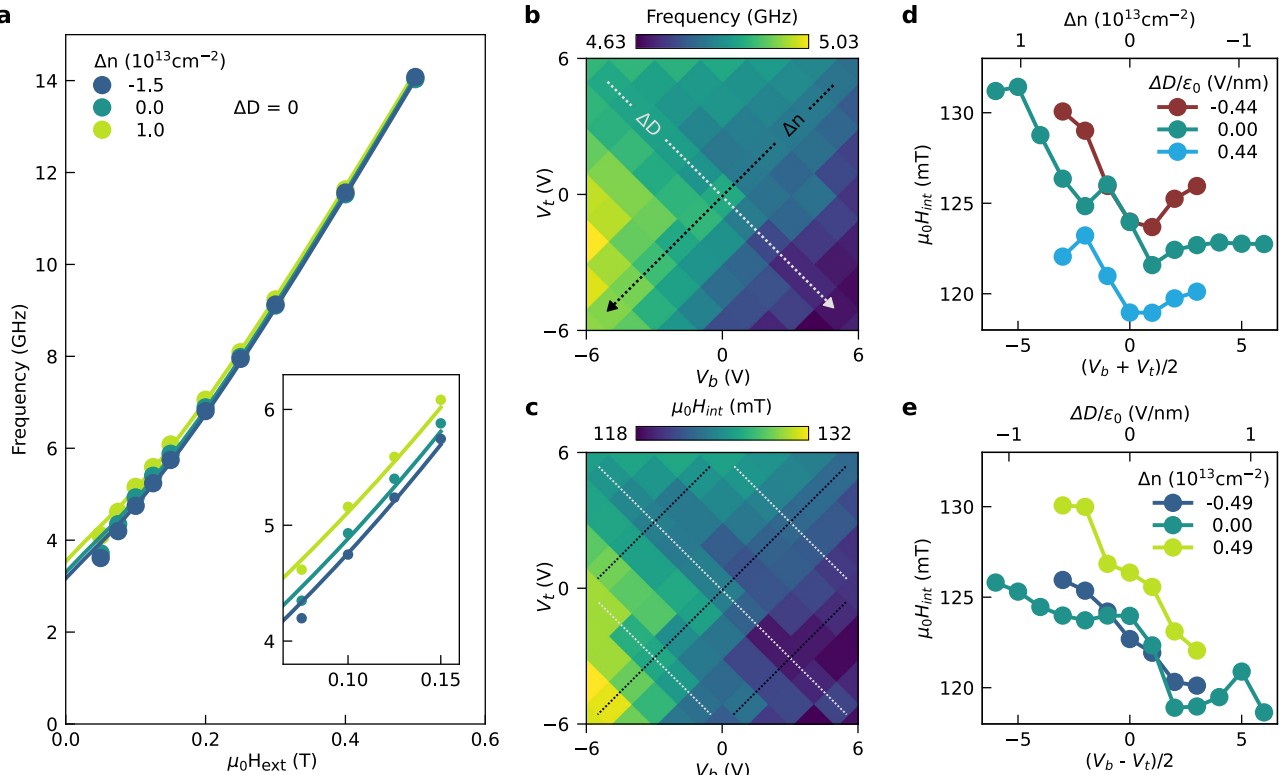

**Fig. 2 | Gate-dependence of precession frequency and internal effective field.**
**a** Frequency of oscillations as a function of external magnetic field, for different
values of $\Delta n$. The circles are the frequencies extracted from the TRFE data for
$\Delta t > 26$ ps. Solid lines are best fits of Eq (1). *Inset*: Close-up of the data for low fields,
showing the frequency shift due to gating. The error bars are smaller than the
markers. **b** Frequency of the oscillations in the TRFE signal at $\mu_0 H_{ext} = 100$ mT for
various values of top ($V_t$) and back gate voltages ($V_b$). The black and gray arrows
indicate, respectively, the directions of constant $\Delta D$ and varying $\Delta n$, and of con-
stant $\Delta n$ and varying $\Delta D$. For other values of $H_{ext}$ see Supplementary Fig. 10.
**c** Internal effective field as a function of $V_t$ and $V_b$. **d, e** The dependence of the
internal effective field on $\Delta n$ for fixed $\Delta D$ (**d**) and on $\Delta D$ for fixed $\Delta n$ (**e**), with solid
lines to guide the eye. The traces are taken along the dotted lines indicated in (**c**).

charge carriers along the thickness of the CGT flake, leading to $\Delta n$-
induced local changes in the magnetization parameters.

Typical results from the time-resolved Faraday ellipticity (TRFE)
measurements for different values of $\Delta n$, with $\Delta D = 0$, are shown in
Fig. 1d. For $\Delta t < 0$ the signal is constant, since the magnetization is at its
steady state value. All traces show a sharp increase at $\Delta t = 0$, indicating
a fast laser-induced dynamics. For $\Delta t > 0$, the TRFE traces show oscil-
lations, indicating a precession of the magnetization induced by the
pump pulse.

We observe that the magnetization dynamics strongly depends
on $\Delta n$, with the amplitude, frequency and starting phase of the
oscillations in the TRFE signal all being affected. The most striking
observation is that the amplitude of the TRFE signal increases by more
than a factor of seven when $\Delta n$ is changed from $1.0 \times 10^{13}$ cm$^{-2}$
to $-1.5 \times 10^{13}$ cm$^{-2}$. The observations of modulation of both the
amplitude and starting phase of the oscillations hint at a change in the
pump excitation process. The change in oscillation frequency due to
$\Delta n$ is better visible in the Fourier transform of the signals, shown in
Fig. 1e (see Methods for details on the Fourier transform). This ana-
lysis clearly shows that both the frequency and amplitude of the
magnetization precession are tuned by $\Delta n$. All these observations
point to an effective control of the (dynamic) magnetic properties of
CGT by electrostatic gating. We note that the photo-induced charge
carrier density from the pump pulse is of the same order of magni-
tude as the gate-induced $\Delta n$. However, as discussed in Supplementary
Note 7, this does not affect our conclusions about the effect of elec-
trostatic gating.

The origin of the electric control of the magnetization dynamics
can be further understood by analyzing the precession frequency at
various magnetic fields and values of $\Delta n$ (Fig. 2a). For magnetic fields

below 250 mT we observe a significant shift of the frequency (4–10%)
by changing the charge carrier density. This is clearly visible in the inset
of Fig. 2a, which shows a close-up of the data up to 150 mT. The change
in precession frequency for different values of $\Delta n$ strongly points
towards a modulation of the magnetization parameters of CGT as a
function of the Fermi level, controlled by $\Delta n$.

A quantitative analysis of the oscillation frequency ($f$) as a function
of $H_{ext}$ can be used to extract the magnetization dynamics parameters
of the device. Our data is well described by the ferromagnetic reso-
nance mode obtained from the Landau–Lifshitz–Gilbert (LLG) equa-
tion with negligible damping[61]:

$$f = \frac{g\mu_B\mu_0}{2\pi\hbar}\sqrt{|\mathbf{H}_{eff}|\left(|\mathbf{H}_{eff}| - H_{int}\sin^2(\theta_M)\right)}, \quad (1)$$

where $g$ is the Landé g-factor, $\mu_B$ the Bohr magneton,
$\mathbf{H}_{eff} = \mathbf{H}_{ext} + H_{int}\cos(\theta_M)\hat{z}$, with $H_{int} = 2K_u/(\mu_0 M_s) - M_s$, $M_s$ the saturation
magnetization, and $\theta_M$ the angle between $\mathbf{M}$ and the sample normal (z-
direction). Note that in ref. 61, $|\mathbf{H}_{eff}|$ is expressed as $H_{ext}\cos(\theta_H -
\theta_M) + H_{int}\cos^2(\theta_M)$ (see Supplementary Note 4). The angle $\theta_M$ is cal-
culated by minimizing the magnetic energy in the presence of an
external field, perpendicular magnetic anisotropy, and shape
anisotropy[46]. We obtain the g-factor and $H_{ext}$ of the CGT by fitting the $f$
versus $H_{ext}$ data (e.g. the data presented in Fig. 2a) using Eq. (1), as
explained in the Methods. We note that for $|\mu_0 H_{ext}| < 50$ mT, the
magnetization is likely not completely saturated, since laser excita-
tion can lead to multi-domain formation[62,63], which affects the magnetiza-
tion dynamics. The fitting yields $g \approx 1.89$ with no clear dependence on
$\Delta n$ or $\Delta D$, which is in agreement with (albeit slightly lower than) the
values reported for CGT[40,42,46] (see Supplementary Notes 5 and 6 for

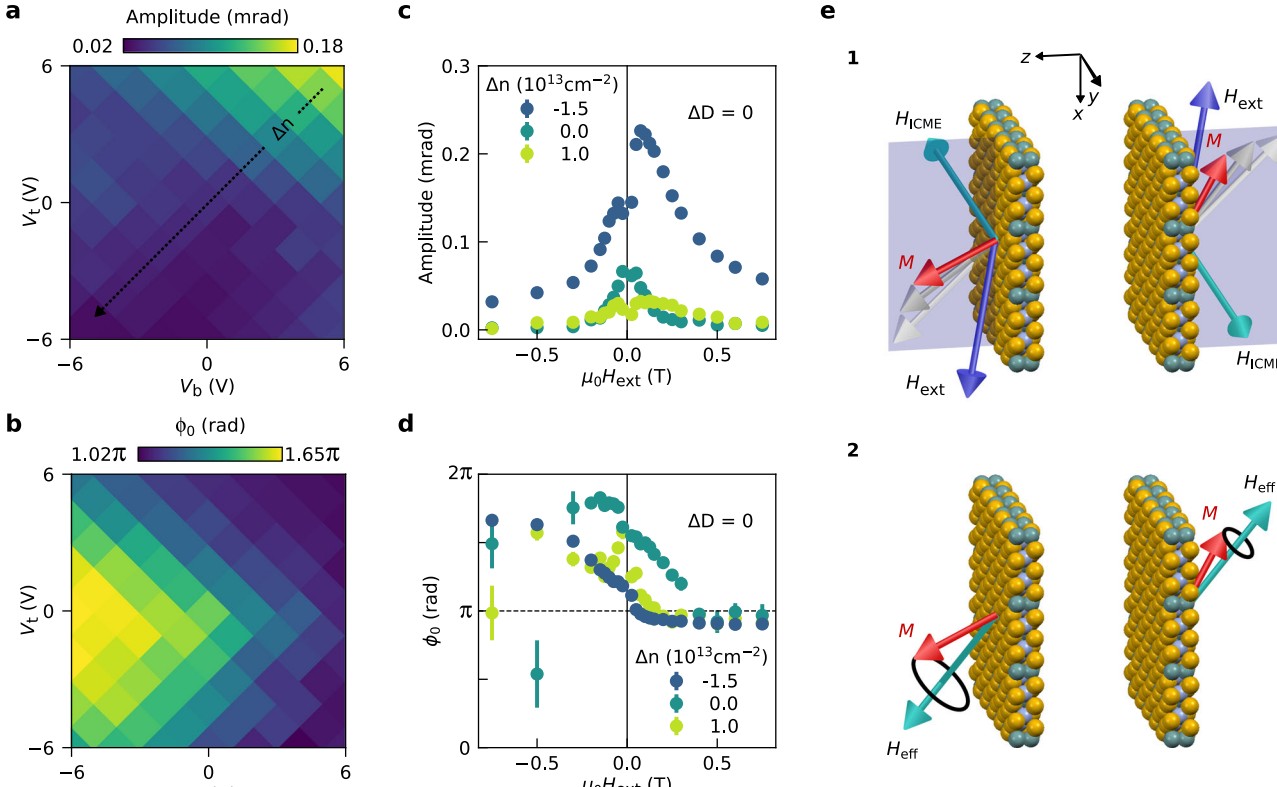

**Fig. 3 | Gate dependence of magnetization precession amplitude and phase. a, b** Gate dependence of the amplitude (**a**) and starting phase (**b**) of the oscillations in the TRFE measurements at $\mu_0 H_{ext} = 100$ mT. For other values of $H_{ext}$ see Supplementary Figs. 8 and 9. **c, d** External magnetic field dependence of the amplitude (**c**) and starting phase (**d**) of the oscillations for different values of $\Delta n$ at $\Delta D = 0$. The values are extracted from the TRFE data for $\Delta t > 26$ ps. The error bars indicate the standard deviation obtained from the least squares fitting procedure. **e**, Schematics of the inverse Cotton-Mouton effect for opposite directions of $H_{ext}$. The magnetization direction is depicted by a red arrow, the external magnetic field in blue, the effective magnetic field induced by the ICME and the effective field are shown in cyan. The $xz$-plane is highlighted by the shaded region.

more details). We also find no clear dependence of the precession damping time ($\tau_{osc}$) on $\Delta n$ or $\Delta D$. The intrinsic Gilbert damping we obtain from our measurements is about $6 \times 10^{-3}$ (see Supplementary Note 8), in line with values found in literature[41,46].

The internal effective field shows a clear dependence on both $V_t$ and $V_b$, as shown in Fig. 2c, with values similar to the ones found in other studies[46]. Upon comparing Fig. 2c to b, one notices that the gate dependence of $H_{int}$ is very similar to that of the precession frequency at $\mu_0 H_{ext} = 100$ mT. This suggests that the gate dependence of the precession frequency is caused by the gate-induced change in $H_{int}$. From the dependence of $H_{int}$ on $V_t$ and $V_b$, we extract its behavior as a function of $\Delta n$ and $\Delta D$, shown in Fig. 2d and e. We observe that $H_{int}$ decreases with both increasing $\Delta n$ and $\Delta D$. The dependence of $H_{int}$ on $\Delta n$ is consistent with theoretical calculations[60], showing that $K_u$, and therefore $H_{int}$, is reduced upon increasing the electron density in the same order of what we achieve in our sample. The $\Delta n$ dependence of $H_{int}$ is also consistent with the dependence of the coercive field obtained from static measurements (see Supplementary Note 3), providing further evidence that the change in $f$ is driven by a change in $K_u$.

Now we draw our attention to the large modulation of the oscillations in the TRFE measurements with varying gate voltage, as shown in Fig. 1d and e. Here we attribute this change in magneto-optical signal amplitude to an actual increase in amplitude of the magnetization precession (increase in the precession angle) and not to an increase in the strength of the Faraday effect. This is supported by our observation that the time-resolved measurements for different combinations of gate voltages are not simply scaled – i.e. the amplitude of the oscillations and their (ultrafast demagnetization) background scale

differently. A detailed discussion can be found in Supplementary Note 9.

Figure 3a clearly shows that the magnetization precession amplitude is mostly affected by $\Delta n$, and to a much lesser extent by $\Delta D$. The precession amplitude versus $H_{ext}$ for various values of $\Delta n$ is presented in Fig. 3c. As noted before, for $|\mu_0 H_{ext}| < 50$ mT the magnetization is likely not completely saturated, and therefore the magnetization dynamics can deviate from the general trend. We find that not only the precession amplitude for a given $H_{ext}$ is strongly modulated by $\Delta n$, but its decaying trend with $H_{ext}$ is also strongly affected. Additionally, we observe another interesting effect: the amplitude shows an asymmetry in the sign of the applied magnetic field, which is also dependent on $\Delta n$. This latter is unexpected, especially since the observed precession frequency is symmetric in $H_{ext}$ (see Supplementary Note 11). A similar precession frequency for opposite magnetic fields indicates that the magnetocrystalline anisotropy and the saturation magnetization are independent of the sign of $H_{ext}$. Therefore, we conclude that the origin of the modulation of the precession amplitude is related to the excitation mechanism of the magnetization precession (see Supplementary Note 12 for the complete discussion).

To get further insight into the microscopic mechanisms involved in the optical excitation of magnetization dynamics, we analyze the magnetic field dependence of the starting phase ($\phi_0$) of the precessions for different values of $V_t$ and $V_b$ (Fig. 3b). Unlike the amplitude, we find that $\phi_0$ depends on both $\Delta n$ and $\Delta D$. As can be seen in Fig. 3d, the behavior of $\phi_0$ with $H_{ext}$ is also modulated by $\Delta n$. For a purely thermal excitation of the magnetization dynamics one would expect $\phi_0(-H_{ext}) = \pi + \phi_0(H_{ext})$ in our geometry. Nonetheless, we observe that

$\phi_0$ for positive and negative magnetic fields differ by less than $\pi$. Moreover, $\Delta n$ seems to also affect the trend on how $\phi_0$ approaches the values at high magnetic fields. Combined with the observed asymmetry of the precession amplitude, our data strongly suggests that the optical excitation of the magnetization dynamics is not dominated by a thermal excitation ($\Delta K$ mechanism) as previously reported for other van der Waals magnets[44–48,53].

## Opto-magnetic effects

Coherent opto-magnetic mechanisms provide possible alternatives for the optical excitation of magnetization dynamics in CGT. Here we find that our data can be explained by two of these mechanisms that are compatible with a linearly-polarized pump pulse, the inverse Cotton-Mouton effect (ICME)[64–66] and photo-induced magnetic anisotropy (PIMA)[67], in addition to the conventional (thermal) $\Delta K$ mechanism[68]. The ICME, which could be described by impulsive stimulated Raman scattering, relies on the generation of an effective magnetic field upon interaction with linearly polarized light in a magnetized medium[64–66,69,70]. This effective magnetic field is proportional to both the light intensity and magnetization. For pulsed laser excitation, the ICME generates a strong impulsive change in $H_{ext}$ that results in a fast rotation of the magnetization. Therefore, this effect can cause the amplitude of the precession to be asymmetric in $H_{ext}$[66,70].

Figure 3 e illustrates how the ICME could result in an asymmetric magnetic field dependence of the amplitude. For simplicity, we only consider the $y$-component of the generated effective magnetic field, since this component is responsible for the asymmetry. (1) A sample with perpendicular magnetic anisotropy is subject to an external magnetic field $\mathbf{H}_{ext}$ ($-\mathbf{H}_{ext}$) in the $xz$-plane, pointing in the positive (negative) direction of both axes. In equilibrium, the magnetization points along the total effective field, as indicated by the light gray arrow. During laser pulse excitation, the ICME results in an effective magnetic field along the $y$-axis, rotating the magnetization either towards the $z$-axis or the $x$-axis, depending on the sign of $\mathbf{H}_{ext}$. Additionally, the ultrafast demagnetization process leads to a reduction of the magnetization. (2) After the laser pulse, the magnetization precesses around the total effective field that is composed of $\mathbf{H}_{ext}$ and $\mathbf{H}_{int}$. Depending on the sign of $\mathbf{H}_{ext}$, the ICME has either rotated $\mathbf{M}$ towards or away from $\mathbf{H}_{eff}$, resulting in different precession amplitudes.

The second coherent mechanism for laser-induced magnetization dynamics is PIMA, which leads to a step-like change in $\mathbf{H}_{eff}$ due to pulsed laser excitation[66]. This mechanism has been reported to arise from an optical excitation of nonequivalent lattice sites (e.g. dopants and impurities), which effectively redistributes the ions and hence changes the magnetic anisotropy[71–73]. Unlike the ICME, the PIMA mechanism is not expected to lead to an asymmetry of the magnetization precession amplitude upon a reversal of $H_{ext}$, because it is present for times much longer than the period of precession and therefore acts as a constant change of the effective magnetic field[67,72–74].

All three discussed mechanisms for inducing magnetization precession—ICME, PIMA and the $\Delta K$ mechanism—are affected by electrostatic gating. The opto-magnetic effects can be affected through a change in e.g. the polarization-dependent refractive index and the occupation of charge states of ions and impurities. Additionally, the $\Delta K$ mechanism can be affected by the changes in charge relaxation pathways through, for example, electron–electron and electron–phonon interactions. We find that the combination of the above mechanisms can describe quantitatively the starting phase and qualitatively the amplitude of the observed magnetic field dependence shown in Fig. 3 (see Supplementary Note 13). The balance between these three mechanisms affects the magnetic field dependence of the amplitude and the starting phase, increases or decreases the asymmetry in the induced precession amplitude, and changes the steepness

of the starting phase versus magnetic field graph. Therefore, since our data shows a change in these properties, we conclude that the relative strength of the mechanisms for excitation of magnetization precession are effectively controlled by electrostatic gating.

We envision that the electric control over the optically-induced magnetization precession amplitude demonstrated here can be applied to devices which make use of spin wave interference for signal processing[12–14]. This should lead to an efficient electric control over the mixing of spin waves, leading to an easier on-chip implementation of combined magnonic and photonic circuits. Even though the control over the precession frequency shown here is still modest ($\approx$10%), we believe it can be further enhanced by the use of more effective electrostatic doping, such as using high-$\kappa$ dielectrics or ionic-liquid gating which is capable of achieving over one order of magnitude higher changes in carrier densities than the ones reported here[21,24,25,32,75]. Using thinner CGT would further increase the gating efficiency since the gate-induced charges are confined to a smaller volume. We note that due to the non-monotonic behavior of the magnetic anisotropy energy with changes in charge carrier density, one might expect more drastic changes on $H_{int}$ for larger changes in $\Delta n$. This control over the magnetic anisotropy can then be used for the electrostatic guiding and confinement of spin waves, leading to an expansion of the field of quantum magnonics. Finally, the presence of coherent optical excitation of magnetization dynamics we observed in CGT should also lead to a more energy-efficient optical control of magnetization[76]. Therefore, the electric control over magnetization dynamics in CGT shown here provides the first steps towards the implementation of vdW ferromagnets in magneto-photonic devices that make use of spin waves to transport and process information.

## Methods

### Sample fabrication

The thin hBN and graphite flakes are exfoliated from bulk crystals (HQ graphene) on an oxidized silicon wafer (285 nm oxide thickness). The CGT flakes are exfoliated in the same way in an inert (nitrogen gas) environment glove box with less than 0.5 ppm oxygen and water to prevent degradation. The flakes are selected using optical contrast and stacked using a polycarbonate/polydimethylsiloxane stamp by a dry transfer van der Waals assembly technique[77]. First, an hBN flake (21 nm thick) is picked up, followed by the CGT flake (10 nm thick). Next, a thin graphite flake is picked up to make electrical contact with a corner of the CGT, and extends beyond the picked-up hBN flake. After this, a second hBN flake (20 nm thick) is picked up and a thin graphite flake to function as the back gate electrode. This stack is then transferred to an optically transparent fused quartz substrate finally a thin graphite flake is transferred on top the stack to function as the top gate electrode. The device is then contacted by Ti/Au (5/50 nm) electrodes fabricated using conventional electron-beam lithography and thin metallic film deposition techniques.

### Measurement setup

All measurements are done at 10 K under low-pressure (20 mbar) Helium gas. The sample is mounted at an angle, such that the sample normal makes an angle of 50 degrees with the external magnetic field and the laser propagation direction.

The ~200 fs long laser pulses are generated by a mode-locked Ti:Sapphire oscillator (Spectra-Physics MaiTai), at a repetition rate of 80 MHz. After a power dump, the pulses are split in an intense pump and a weaker probe pulse by a non-polarizing beam splitter. The pump beam goes through a mechanical delay stage, allowing us to modify the time delay between pump and probe by a change in the optical path length. To allow for a double-modulation detection[54,55], the pump beam goes through an optical chopper working at 2173 Hz. The polarization of the pump is set to be horizontal (p-polarized with respect to the sample), to allow us to block the pump beam through a

polarization filter at the detection stage. The initially linearly polarized probe pulse goes through a photoelastic modulator (PEM) which modulates the polarization of the light at 50 kHz. A non-polarizing beam splitter is used to merge the pump and probe beams on parallel paths, with a small separation between them. From here, they are focused onto the sample by an aspheric cold lens with a numerical aperture of 0.55. The probe spot size (Full Width at Half Maximum) is ~1.8 $\mu m$ and the pump spot size is ~3.4 $\mu m$, both elongated by a factor of $1/\sin(50°)$ because the laser hits the sample at 50° with respect to the sample normal. The fluence of the pump and probe pulses are $F_{pump} = 25\ \mu J/cm^2$ and $F_{probe} = 5.7\ \mu J/cm^2$, respectively. The transmitted light is collimated by an identical lens on the opposite side of the sample and leaves the cryostat. The pump beam is blocked and the probe beam is sent to a detection stage consisting of a quarter wave plate, a polarization filter, and an amplified photodetector. The quarter wave plate and the polarization filter are adjusted until they compensate for the change in polarization caused by the optical components between the PEM and the detection stage, ensuring that our signals are purely due to the rotation or ellipticity of the probe polarization induced by our samples. The first and second harmonic of the signal (50 or 100 kHz) obtained at the photodetector, which can be measured using lock-in amplifiers, are then proportional to the change in ellipticity and rotation respectively due to the Faraday effect of the sample. For the time-resolved Faraday ellipticity measurements, we use one lock-in amplifier to measure the signal corresponding to the first harmonic of the PEM at the photodetector output, and send the output of this lock-in amplifier to the input of the next lock-in amplifier, which is referenced to the frequency of the chopper. The output of the second lock-in amplifier is the pump-induced change in the Faraday ellipticity, which we call the TRFE signal for short. The sign is chosen such that a positive TRFE signal corresponds to a decrease in magnetization. For static magneto-optic Faraday effect measurements we have blocked the pump beam before reaching the sample, and we measure the first and second harmonic of the PEM in the photodetector signal simultaneously.

## Calculating $\Delta n$ and $\Delta D$ from the gate voltages

The gate-induced change in charge carrier density ($\Delta n$) and displacement field ($\Delta D$) at the CGT are calculated from the applied gate voltages using a parallel plate capacitor model. The displacement field generated by the top ($D_t$) and back ($D_b$) gates is given by $D_i = \varepsilon_{hBN} E_i = \frac{1}{2}\sigma_{free,i}$, where $i$ denotes $t$ or $b$, $\varepsilon_{hBN} = 3\varepsilon_0$ is the hBN dielectric constant[78,79] with $\varepsilon_0$ the vacuum permittivity, and $\sigma_{free}$ the free charge per unit area. The applied top and back gate voltages are related to $\sigma_{free}$ by $V_i = -\int D_i/\varepsilon\,dz$. This equation, combined with the condition of charge neutrality, gives the following three relations:

$$V_t/d_t = \frac{\sigma_t - \sigma_{CGT} - \sigma_b}{2\varepsilon_{hBN}},$$
$$V_b/d_b = \frac{\sigma_b - \sigma_{CGT} - \sigma_t}{2\varepsilon_{hBN}},$$
$$0 = \sigma_t + \sigma_b + \sigma_{CGT},$$

where $d_{t,b}$ denotes the thickness of the top (21 nm) and bottom (20 nm) hBN flakes, and $\sigma_i$ the free charge per unit area in the top gate ($t$), back gate ($b$), and the CGT flake. Solving this set of equations yields:

$$\sigma_t = \varepsilon_{hBN} V_t/d_t,$$
$$\sigma_b = \varepsilon_{hBN} V_t/d_t,$$
$$\Delta n = \sigma_{CGT}/e = -\frac{\varepsilon_{hBN}}{e}\left(\frac{V_t}{d_t} + \frac{V_b}{d_b}\right),$$

where $e$ is the positive elementary charge. Note that for positive gate voltages, a negative charge carrier density is induced in the CGT. For the gate-induced change in the displacement field at the CGT layer, we get:

$$\Delta D = (\sigma_b - \sigma_t)/2 = -\varepsilon_{hBN}(V_t/d_t - V_b/d_b)/2$$

Filling in the values for the thickness of the hBN flakes and dielectric constant of hBN gives $\Delta D/\varepsilon_0$ and $\Delta n$ at the CGT:

$$\Delta n = -(0.79 V_t + 0.83 V_b) \times 10^{12}\,V^{-1}cm^{-2}$$
$$\Delta D/\varepsilon_0 = -(0.071 V_t - 0.075 V_b)\,nm^{-1}.$$

Filling in the maximum gate voltage we can apply to our sample, 9 V for $V_{top}$ and $V_{back}$, yields $\Delta n = -1.5 \times 10^{13}\,cm^{-2}$. This is a typical value for a dual-gate geometry with hBN as a dielectric[23,53]. Throughout the main text we use $\Delta D/\varepsilon_0$ instead of $\Delta D$ for easier comparison of our values of the gate-induced change in the displacement field with values mentioned in other works. Note that we use the conversion factor $\varepsilon_0$ and not the permittivity of CGT. Therefore, the values for the $\Delta D$ that we report are the equivalent electric field values in vacuum, not in CGT.

## Windowed Fourier transform

The frequency spectra of the TRFE oscillations shown in Fig. 1e are calculated from the TRFE measurements using a windowed Fourier transform. The type of window used for this calculation if the Hamming window, which extends from $\Delta t = 0$ up to the last data point. The Fourier amplitude spectrum ($A(f)$) of the TRFE oscillations is calculated as

$$A(f) = \left(\sum_{\Delta t > 0}\left[W_{Ham}(\Delta t)y(\Delta t)\sin(2\pi f\Delta t)\right]^2 + \left[W_{Ham}(\Delta t)y(\Delta t)\cos(2\pi f\Delta t)\right]^2\right)^{1/2},$$

where $W_{Ham}$ is the Hamming window, $y$ the data points of the TRFE measurements, and $f$ the frequency.

## Determining the $g$-factor and $H_{int}$

The Landé $g$-factor and $H_{int}$ can be extracted by fitting the magnetic field dependence of the precession frequencies with Eq. (1). The values of $g$ and $H_{int}$ we obtained from the fit were, in most cases, strongly correlated. Therefore, we first determined $g$ by fitting the data for $\mu_0 H_{ext} \geq 125$ mT, since $g$ is most sensitive to the slope at high fields. This yields $g = 1.89 \pm 0.01$. If we further allow for an additional uncertainty in the mounting angle of the sample, the g-factor can change by ~0.1. Then we determine $H_{int}$ by fitting Eq. (1) for all remaining measurements fixing $g = 1.89$. We note that the values for $H_{int}$ do depend on the exact value of $g$, but the modulation due to electrostatic gating is not affected, as is shown in Supplementary Note 5.

## Extracting the magnetization precession parameters

We extract the amplitude, frequency, and starting phase of the oscillations in the TRFE measurements by fitting the data for $\Delta t > 26$ ps with the phenomenological formula[46,61]

$$y = y_0 + ae^{-\Delta t/\tau_{osc}}\cos(2\pi f\Delta t - \phi_0) + A_l e^{-\Delta t/\tau_l} + A_s e^{-\Delta t/\tau_s}. \quad (2)$$

This formula describes a phase-shifted sinusoid on top of a double exponential background. The background captures the demagnetization and remagnetization of the CGT, while the sinusoid describes the magnetization precession.

## Data availability

The raw data generated in this study and the processed data have been deposited in the Zenodo data base under accession code https://doi.org/10.5281/zenodo.8321758[80].

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

## Acknowledgements

The authors thank Bart J. van Wees for critically reading the manuscript and providing valuable feedback, and we thank J. G. Holstein, H. Adema, H. de Vries, A. Joshua and F. H. van der Velde for their technical support. This work was supported by the Dutch Research Council (NWO) through grants STU.019.014 and OCENW.XL21.XL21.058, the Zernike Institute for Advanced Materials, the research program "Materials for the Quantum Age" (QuMat, registration number 024.005.006), which is part of the Gravitation program financed by the Dutch Ministry of Education, Culture and Science of Education, Culture and Science (OCW), and the European Union (ERC, 2D-OPTOSPIN, 101076932). Views and opinions expressed are however those of the author(s) only and do not necessarily reflect those of the European Union or the European Research Council. Neither the European Union nor the granting authority can be held responsible for them. The device fabrication and nanocharacterization were performed using Zernike NanoLabNL facilities.

## Author contributions

M.H.D.G. conceived and supervised the research. F.H. designed and fabricated the samples, performed the measurements, analyzed the data, and calculated the effect of coherent excitations on the magnetization precession under M.H.D.G. supervision. F.H. and R.R.R.L. built and tested the measurement setup. F.H., M.H.D.G., and B.K. discussed the data and provided the interpretation of the results. F.H. and M.H.D.G. co-wrote the manuscript with input from all authors.

## Competing interests

The authors declare no competing interests.
