## [Peer Review File · Nature Communications]

Reviewers' Comments:

Reviewer #1:

Remarks to the Author:

In this manuscript, the authors studied gate control of the coherent magnon frequency and amplitude in a thin ferromagnetic semiconductor Cr₂Ge₂Te₆. The coherent magnons were excited and probed by time-resolved Faraday ellipticity measurements. They found that the magnon frequency and amplitude depend on both the carrier concentration and the electric displacement field. In addition, they found that the excitation of magnetization dynamics is mainly due to coherent opto-magnetic effects. I think the results are novel and interesting, demonstrating that magnons in Cr₂Ge₂Te₆ can be potentially used in ultrafast opto-magnonic devices. The authors are suggested to address the following questions before publishing the paper.

1. The thickness of the Cr₂Ge₂Te₆ is not mentioned in the main text or Methods. One needs to find this information in the Supplementary material.
2. The thickness of the sample is ~10 nm (~14 layers), which is not in the 2D limit (c.f. Ref. 2). Have the authors studied even thinner samples? Will the magnons or their tunability be different in the thinner samples?
3. Have the authors studied temperature dependence of the gate controlled magnon properties?
4. Both static and time-resolved Faraday ellipticity signals are shown in arb. units. It is important to know the absolute values, which are easy to obtain from the measurements, so that the results can be compared with previous studies and also potentially future work.
5. There is a recent study about coherent magnon dynamics in few-layer MnBi₂Te₄ using time-resolved optical spectroscopy (arxiv: 2304.09390). It is very relevant to cite this work.
6. Do the authors know the intrinsic carrier density of the sample? N type or p type?
7. Can the authors estimate the photoinduced carrier concentration and plot it on top of Fig. S6? It seems that the photodoping may be comparable to or even bigger than the electrically induced density.
8. Please check Eq. (1), which looks different from the Kittel formula (e.g., Lattery, D., Zhu, J., Huang, D.B., Wang, X.J., "Ultrafast Thermal and Magnetic Measurements Enabled by Time-Resolved Magneto-Optical Kerr Effect," Chapter 9).
9. The fitting of the resonant frequency at low external magnetic field is not very well (e.g., Fig. 2a), where the internal field dominates. Can the authors comment on this?
10. Please pay attention to some grammar issues and typos.

Reviewer #2:

Remarks to the Author:

In this work, the authors study magnetization dynamics in a Cr₂Ge₂Te₆ (CGT)-based heterostructure. Using the dual gate geometry, they can independently control charge carrier density and the perpendicular electric field. Based on their time-resolved Faraday ellipticity measurements, they claim that the magnetization dynamics can be controlled by the gate, in particular, the microscopic mechanisms involve the optical excitation of magnetization dynamics.

First of all, electric control magnetization has been extensively studied in 2D magnets. Even for the optical excitation of magnetization (i.e., non-thermal effect), it has been demonstrated in both metallic (Fe₃GeTe₂, see PRL 125, 267205(2020)) and semiconducting (NiPS₃, see Sci. Adv. 7(23) eabf3096 (2021)) materials. Therefore, I cannot agree this topic "remains to be explored".

Moreover, the CGT sample has been investigated using the same pump-probe technique in APL 116, 223103(2020). In that paper, the magnetization dynamics can be largely attributed to the pump-induced thermal effect. In this manuscript, if the claim of optical excitation of magnetization is true, it may be considered as noteworthy results. But, I think it is necessary to inspect this claim with more prudence, especially when the claim is mainly based on the observation of the asymmetric precession amplitude with H_{ext}. I think the authors need to rule out the possibility of artifacts due to the asymmetric sample configuration or the detection efficiency.

Reviewer #3:

Remarks to the Author:

The manuscript by Hendriks et al. reports the magnon oscillations in Cr₂Ge₂Te₆, and how does the amplitude and phase of excited magnons change as a function of the electrostatic doping and electric field. The measured magnon oscillations showed consistent magnetic field dispersion compared to the LLG equations. The observed doping and electrical field are also novel and can raise broad interest in the first of spintronics and 2D materials. There are a few questions/comments about the experimental interpretation of the inverse Cotton-Mouton effect and the extraction of doping. The authors will need to first address these issues before I recommend its publication in Nature Communications.

1. The TRFE shows a positive increase after time-zero. Is this the expected sign of a demagnetization? The data sign can be changed depending on the chosen phase, but does this correspond to an increase or decrease in overall magnetization?
2. The device top and bottom gate has 20nm hBN. The extracted doping level is on the scale of $2 \times 10^{13}/\text{cm}^2$. The extracted doping seems too large given the voltages applied (e.g. in Fig. 2b&c). Please double-check your estimation with other publications, e.g. in <https://pubs.acs.org/doi/full/10.1021/acs.nanolett.7b03667>
3. Given the large tilting angle of the sample, which gives rise to external field anisotropy, I think it will make more sense to plot Fig. 3c&d with the effective magnetic field (or other fields that take into account of the sample tilting angle), instead of the external field. It's otherwise unclear what is the impacts of the sample tilting for this field anisotropy discussion.
4. To have sizable ICME effect, the field strength of pulsed laser must be large. Give the laser fluence in this experiment, do the authors estimate a large enough ICME to give rise to any observable effects? ICME should also have a specific fluence dependence. In the supplementary figure, the fluence dependence in Fig. S5 seems to indicate a relatively small effect on the magnon amplitude and phase.

Reviewer #1

In this manuscript, the authors studied gate control of the coherent magnon frequency and amplitude in a thin ferromagnetic semiconductor $\text{Cr}_2\text{Ge}_2\text{Te}_6$. The coherent magnons were excited and probed by time-resolved Faraday ellipticity measurements. They found that the magnon frequency and amplitude depend on both the carrier concentration and the electric displacement field. In addition, they found that the excitation of magnetization dynamics is mainly due to coherent opto-magnetic effects. I think the results are novel and interesting, demonstrating that magnons in $\text{Cr}_2\text{Ge}_2\text{Te}_6$ can be potentially used in ultrafast opto-magnonic devices. The authors are suggested to address the following questions before publishing the paper.

We are happy to hear that the reviewer finds our work interesting for the broader research community and appreciates its novelty.

- 1. The thickness of the $\text{Cr}_2\text{Ge}_2\text{Te}_6$ is not mentioned in the main text or Methods. One needs to find this information in the Supplementary material.*

We thank the reviewer for pointing this out. We now mention the thickness of the $\text{Cr}_2\text{Ge}_2\text{Te}_6$ flake in the main text and in the Methods section.

Changes to the manuscript: In the main text, section 2, first paragraph, we included: “... *consists of a 10 nm thick CGT flake ...*”; In the Methods section (section 6.1) we have added: “... *by the CGT flake (10 nm thick).*”

- 2. The thickness of the sample is ~10 nm (~14 layers), which is not in the 2D limit (c.f. Ref. 2). Have the authors studied even thinner samples? Will the magnons or their tunability be different in the thinner samples?*

The reviewer correctly points out that the thickness of our sample (10 nm) is not in the atomically-thin limit. However, it is thin enough to obtain a large field-effect on the magnetic properties, which was our main goal. We have not studied the thickness dependence of the electric tunability of the effects we show. Nonetheless, we expect that the magnon tunability will be enhanced further when the CGT becomes thinner, since the electrically induced carriers will be distributed in a smaller volume. We believe that this should be the focus of future work which aims to optimize the electrical tuning of magnons in this material. We have added a note about this in the conclusion.

Changes to the manuscript: In the main text, conclusions: “*Using thinner CGT would further increase the gating efficiency since the gate-induced charges are confined to a smaller volume.*”

- 3. Have the authors studied temperature dependence of the gate controlled magnon properties?*

We have not studied the temperature dependence of the gate controlled magnon properties.

We expect that at higher temperatures the magnetic resonance frequency would decrease, since both the saturation magnetization and the magnetocrystalline anisotropy decrease with increasing temperature. We would also expect the TRFE signal, and therefore the signal to noise ratio, to decrease because the saturation magnetization decreases. We do not know how the temperature

would affect the gate control of the magnon properties. Since the temperature would add yet another parameter space to be explored, we decided to focus our studies to lower temperatures only.

4. *Both static and time-resolved Faraday ellipticity signals are shown in arb. units. It is important to know the absolute values, which are easy to obtain from the measurements, so that the results can be compared with previous studies and also potentially future work.*

We thank the reviewer for this suggestion. We have revised all the figures in our manuscript to display the magneto-optical signals in absolute units, *mrad* (see the figure below for the modified panels of Fig. 1 and 3). We have kept the frequency spectrum, which is displayed in Figure 1e, in arbitrary units since it shows a windowed Fourier transform of the data shown in Fig 1d, and only serves to qualitatively show that the frequency and amplitude of the TRFE oscillations vary with magnetic field and gate voltage.

Changes to the manuscript: Main text, the panels Fig. 1d, Fig. 3a and Fig. 3c have been updated as shown in Fig. R1 below.

Fig. R1: Revision of Fig. 1 and 3 of the main text, showing the TRFE signals and magnetization oscillation amplitude in units of *mrad*.

5. *There is a recent study about coherent magnon dynamics in few-layer MnBi2Te4 using time-resolved optical spectroscopy (arxiv: 2304.09390). It is very relevant to cite this work.*

We thank the reviewer for pointing out this relevant work that has slipped our attention.

Changes to the manuscript: We have included the reference above in the main text, in the second paragraph of the introduction.

6. *Do the authors know the intrinsic carrier density of the sample? N type or p type?*

We have not carried out electrical (Hall) measurements of our samples since this significantly complicates the device geometry. Nonetheless, the supplier of our CGT crystals (HQ Graphene) states that the crystals are undoped, with levels lower than what they can characterize via Hall measurements. From the experience of one of our collaborators working on electrical transport devices in CGT, we believe that our devices might be slightly n-doped due to residues from the device fabrication, inducing an extrinsic doping in the order of 10^{11} cm^{-2} , much smaller than the electrically induced charge carriers in our studies.

7. *Can the authors estimate the photoinduced carrier concentration and plot it on top of Fig. S6? It seems that the photodoping may be comparable to or even bigger than the electrically induced density.*

From our estimations (see below) the photo-induced carrier densities right after laser excitation ($\sim 1.6 \times 10^{13} \text{ cm}^{-2}$) should indeed be comparable with the ones induced by electrostatic gating. However, we would like to stress that the energy profiles of the two are drastically different. While electrostatic doping changes the overall doping level (background), laser excitation generates high energy electrons and holes, which rapidly decay and thermalize towards lower energies. Since we use ultrashort laser pulses and not continuous wave lasers, the photoinduced carrier density changes over time. Additionally, we would like to point out that while the carrier excitation densities are indeed comparable to the electrostatic doping, we still observe drastic gating effects on the magnetization dynamics. Finally, our TRFE measurements show a very similar response to electrostatic gating for different pump fluences, as shown in Supplementary Fig. 5. Therefore, we are confident that the photoinduced carrier density does not affect our main findings on the effect of gating on the magnetization dynamics properties, and on the mechanisms of optically induced magnetization dynamics in CGT. Since the photoexcited carrier density changes dynamically, both in energy and magnitude, we prefer to not complicate the picture given by the supplementary figure 6 and keep it as is. We hope that the reviewer agrees with this decision.

To estimate the photoinduced carrier concentration in our experiments, we start by calculating the number of photons per laser pulse per square centimeter that arrive at the CGT, which is $1.6 \times 10^{14} \text{ cm}^{-2}$. To estimate the fraction that gets absorbed by the flake, we use the values of the optical conductivity tensor σ of bulk CGT calculated by Fang et. al. [*Phys. Rev. B* **98**, 1254196 (2018)] For a wavelength $\lambda = 870 \text{ nm}$, this is about $(1.2 - 2i) \times 10^{15} \text{ Hz}$ (in Gaussian units). The refractive index is $\sqrt{1 + 4\pi i \sigma / \omega} = \sqrt{1 + 2\pi i \sigma \lambda / c} = 3.8 + 0.95i$. The intensity of light as a function of distance d traveled through the CGT is $I = I_0 e^{-4\pi d / \lambda} = I_0 e^{-0.014d / \text{nm}}$. The absorbed fraction of the intensity for 10 nm of CGT is $1 - I(10 \text{ nm}) / I_0 = 0.14$. Therefore, about 14% of the photons that travel through the CGT get absorbed. Since a part of the photons that arrive at the CGT get reflected, we take the fraction of photons arriving at the CGT that get absorbed to be in the order of 10%. This results in 1.6×10^{13} photons that create electron-hole pairs. Therefore, the photo-induced charge carrier density right after excitation of the pump pulse is of the order of the maximum gate-induced charge carrier density.

Changes to the manuscript: We have added a new section in the Supplementary Information (Section 7) where the calculation is given in full detail. In the main text (Section 3, third paragraph) we have added a note on the photo-induced carriers, and a reference to the Supplementary Section 7.

8. Please check Eq. (1), which looks different from the Kittel formula (e.g., Lattery, D., Zhu, J., Huang, D.B., Wang, X.J., "Ultrafast Thermal and Magnetic Measurements Enabled by Time-Resolved Magneto-Optical Kerr Effect," Chapter 9).

We highly appreciate that the reviewer has taken the effort of carefully checking our equations. The equation shown in our manuscript is completely equivalent to the one in the reference given by the reviewer, but in a more compact form. To clarify this, we explain here how we derived Eq. 1. The derivation has also been added to the supplementary information.

We have derived Eq. 1 (in SI units) from the Landau-Lifshitz-Gilbert equation in the case of negligible damping. The frequency is expressed as $f = \gamma \sqrt{H_1 H_2} / 2\pi$, with:

$$H_1 = H_{ext} \cos(\theta - \theta_H) + H_{K,eff} \cos^2(\theta), \text{ and } H_2 = H_{ext} \cos(\theta - \theta_H) + H_{K,eff} \cos(2\theta).$$

Using the trigonometric identity $\cos(2\theta) = \cos^2(\theta) - \sin^2(\theta)$, it is straightforward to show that:

$$H_1 - H_{K,eff} \sin^2(\theta). \text{ Furthermore, it can be shown that } H_1 \text{ is equal to } |\mathbf{H}_{eff}|, \text{ by writing } |\mathbf{H}_{eff}| =$$

$$\mathbf{H}_{eff} \cdot \mathbf{H}_{eff} / |\mathbf{H}_{eff}|. \text{ Since the magnetization points along } \mathbf{H}_{eff}, \text{ we have } \mathbf{H}_{eff} / |\mathbf{H}_{eff}| = \mathbf{M} / |\mathbf{M}|.$$

Using this, together with $\mathbf{H}_{eff} = \mathbf{H}_{ext} + H_{int} \cos(\theta) \mathbf{n}$, where \mathbf{n} is the sample normal unit vector, and the dot product identity $\mathbf{u} \cdot \mathbf{v} = |\mathbf{u}| |\mathbf{v}| \cos(\theta_{uv})$, yields $H_1 = |\mathbf{H}_{eff}|$. Using these results, the

expression for the frequency becomes: $f = \gamma / 2\pi \sqrt{|\mathbf{H}_{eff}| (|\mathbf{H}_{eff}| - H_{K,eff} \sin^2(\theta))}$, which is the expression that we use in Eq. 1.

Changes to the manuscript: We have added a new section in the Supplementary Information (Section 4) where this derivation is given in full detail. Main text, right below Eq. 1 we have added a note about the different expressions for f that we use, and a reference to the Supplementary Section 4.

9. The fitting of the resonant frequency at low external magnetic field is not very well (e.g., Fig. 2a), where the internal field dominates. Can the authors comment on this?

We believe that the discrepancy at low values of the external magnetic field is due to the fact that the magnetization of the CGT is not fully saturated after being excited by the pump pulse. As reported by Khela et. al [Nat. Commun. **14**, 1378 (2023)], laser pulses can induce a stripe domain state or stable spin textures in magnetized CGT (reported for a 25 mT magnetic field applied out-of-plane). We expect that the stripe domains and spin textures only form at low magnetic fields, and that for high enough fields (>50 mT) the magnetization of the CGT is uniform. We have added a note and a reference to Khela et. al. in the main text, at the top of page 6 (where Figure 2a is discussed), and at the bottom of page 6 we have changed the note about the discrepancy at low fields in Figure 3c.

Changes to the manuscript: In the main text, we have added: "We note that for ... affects the magnetization dynamics".

In the main text, we have replaced "We note that for ... from the general trend" by "As noted before, ... from the general trend."

10. Please pay attention to some grammar issues and typos.

We would like to thank the reviewer for this comment. We have gone through the whole manuscript once more to improve the grammar and correct typos.

Reviewer #2

In this work, the authors study magnetization dynamics in a $\text{Cr}_2\text{Ge}_2\text{Te}_6$ (CGT)-based heterostructure. Using the dual gate geometry, they can independently control charge carrier density and the perpendicular electric field. Based on their time-resolved Faraday ellipticity measurements, they claim that the magnetization dynamics can be controlled by the gate, in particular, the microscopic mechanisms involve the optical excitation of magnetization dynamics.

First of all, electric control magnetization has been extensively studied in 2D magnets. Even for the optical excitation of magnetization (i.e., non-thermal effect), it has been demonstrated in both metallic (Fe_3GeTe_2 , see PRL 125, 267205(2020)) and semiconducting (NiPS_3 , see Sci. Adv. 7(23) eabf3096 (2021)) materials. Therefore, I cannot agree this topic "remains to be explored".

Moreover, the CGT sample has been investigated using the same pump-probe technique in APL 116, 223103(2020). In that paper, the magnetization dynamics can be largely attributed to the pump-induced thermal effect.

The reviewer is concerned about the novelty of our work, and mentions similarities to other works in literature. We respectfully disagree with the reviewer that the topic we present has been widely explored. To our knowledge, electric control of magnetization dynamics has only been demonstrated in a 2D antiferromagnet (CrI_3 , Ref. 53 in our main text), and the electric control over opto-magnetic phenomena has not been reported before.

The works cited by the reviewer are very distinct from ours. Particularly, in *Phys. Rev. Lett.* **125**, 267205 (2020), a pulsed laser is used to temporarily stabilize the magnetic ordering of the metallic ferromagnet Fe_3GeTe_2 at room temperature. Their main finding is that the magnetic anisotropy energy, saturation magnetization and the Curie temperature change upon optical excitation of carriers. These parameters are all quantified by static Magneto-optic Kerr effect (MOKE) measurements. The authors do not demonstrate electric control over the magnetization and do not explore any magnetization dynamics. Additionally, there is *no coherent opto-magnetic phenomena* reported on this work, with the effects being easily explained by simple photo-excitation of carriers.

In the second work cited by the reviewer, *Sci. Adv.* **7**, eabf3096 (2021), the antiferromagnetic resonance modes in bulk NiPS_3 are investigated by optically pumping the sample at an orbital transition, which subsequently leads to magnetization dynamics. This work is also distinct from ours. The authors study a bulk (3D) antiferromagnetic sample, and do not show any electric gating effects. Even though the pumping of the orbital transition does result on an effect analogous to the optically-induced magnetic anisotropy, there is no sign of the inverse Cotton-Mouton effect, which is key to explain our results.

Finally, we would like to point out that even though optical-pump probe techniques have been employed to study the magnetization dynamics in CGT [*Appl. Phys. Lett.* **116**, 223103 (2020)], here again there is no report on electrostatic control over the magnetization dynamics nor over its excitation. Additionally, as stated before, neither the presence nor control over the opto-magnetic phenomena in CGT has been reported before. The authors of this work can explain all their results through a simple thermal excitation of the magnetization dynamics. However, this is clearly not the case for our findings as we explain below.

In this manuscript, if the claim of optical excitation of magnetization is true, it may be considered as noteworthy results. But, I think it is necessary to inspect this claim with more prudence, especially when the claim is mainly based on the observation of the asymmetric precession amplitude with H_{ext} . I think the authors need to rule out the possibility of artifacts due to the asymmetric sample configuration or the detection efficiency.

We are happy that the reviewer appreciates the novelty of our results on the optical excitation of magnetization dynamics. We would like to point out that we use several pieces of evidence to support our conclusions that opto-magnetic effects play a major role in the excitation of magnetization dynamics in our samples. As the reviewer rightly points out, one signature we find is that the magnetization precession amplitude is asymmetric with respect to the direction of the applied magnetic field. However, we would like to stress that our claims are additionally supported by several different measurements, all reported in the Supplementary Material. We highlight them below.

1. The asymmetry of the magnetization precession amplitude with respect to the direction of the external magnetic field is effectively controlled by electrostatic gating. This cannot be explained by a simple change in the light absorption or heat profile with doping, but is consistent with a change in the opto-magnetic efficiencies.
2. The asymmetry on the initial phase of precession with the direction of the applied magnetic field is *inconsistent* with a thermal (ΔK) mechanism. While for a thermal excitation one expects the phase to differ by π , we find significantly smaller values, which *change upon electrostatic doping*. For a detailed discussion, please see Fig. 3d of the main text and associated discussion, as well as Section 4 of the main text.
3. In our pump-fluence dependence study we show that thermal demagnetization results in an *increase* of the TRFE signals. However, at low pump fluences, when the thermal effects are reduced, we observe a sharp *decrease* of the signal just after excitation. This is inconsistent with a purely thermal excitation. See Supplementary Figure 13 and associated discussion in the Supplementary Information for more detailed arguments.
4. Finally, a simple model including the inverse Cotton-Mouton effect, photo-induced magnetic anisotropy and the thermal ΔK mechanism is capable of *quantitatively* describe the values and asymmetry of the initial precession phase with a change in the direction of the applied magnetic field. Additionally, it also captures the overall trend of the magnetization precession amplitude. We find that the opto-magnetic phenomena are key to reproduce our data. See Section 13 of the Supplementary Information for a detailed discussion.

We have taken great care in considering alternative explanations for our observations, but we could not find another explanation which is consistent with our observations. Therefore, we can only conclude that opto-magnetic effects play an important role in optically-induced magnetization dynamics in CGT. We are confident that our work, showing the first demonstration of electric control over magnetization dynamics in CGT, as well as the presence and control of opto-magnetic phenomena in a 2D magnet will spark the interest of the 2D materials and magnetism communities.

Reviewer #3

The manuscript by Hendriks et al. reports the magnon oscillations in Cr₂Ge₂Te₆, and how does the amplitude and phase of excited magnons change as a function of the electrostatic doping and electric field. The measured magnon oscillations showed consistent magnetic field dispersion compared to the LLG equations. The observed doping and electrical field are also novel and can raise broad interest in the first of spintronics and 2D materials. There are a few questions/comments about the experimental interpretation of the inverse Cotton-Mouton effect and the extraction of doping. The authors will need to first address these issues before I recommend its publication in Nature Communications.

We are happy to see that the reviewer appreciates the novelty of our work and finds our results interesting. We address their comments and questions below.

- 1. The TRFE shows a positive increase after time-zero. Is this the expected sign of a demagnetization? The data sign can be changed depending on the chose phase, but does this correspond to an increase or decrease in overall magnetization?*

We thank the reviewer for asking this question, which points out to us that the explanation of the meaning of the TRFE signals is unclear. We are happy to clarify this point. The TRFE signals are proportional to the pump-induced change of the Faraday ellipticity of the probe. In our definition, for a positive probe Faraday ellipticity, the TRFE is positive for a pump-induced demagnetization. Note that the TRFE changes sign upon reversing the sign of the probe Faraday ellipticity (see Figures 2 and 12 of the Supplementary Material).

To clarify this point, we have modified the caption of Fig. 1 to include a short description of what our TRFE signal is, and have added in the last paragraph of Section 6.2 (Methods, measurement setup) a description on how we measure the TRFE signal using two lock-in amplifiers.

Changes to the manuscript: Main text, Section 6.2, second paragraph: we have added "... at the photodetector, which can be measured using lock-in amplifiers", and "... and rotation respectively ...", and "For the time-resolved ... decrease in magnetization", and "the sample, and ... photodetector signal simultaneously".

- 2. The device top and bottom gate has 20nm hBN. The extracted doping level is on the scale of 2*10¹³/cm². The extracted doping seems too large given the voltages applied (e.g. in Fig. 2b&c). Please double-check your estimation with other publications, e.g. in <https://pubs.acs.org/doi/full/10.1021/acs.nanolett.7b03667>*

The model we use to calculate the induced carrier density in our devices is the same as the one in the reference provided by the reviewer. The main difference in our work is that we have used the dielectric constant for hBN, $\epsilon_{hBN} = 3.8$ [see e.g. Laturia et. al, *npj 2D Mat. and Appl.* **2**, 6 (2018)] while the work cited by the reviewer uses $\epsilon_{hBN} = 2.5$. We would like to highlight that, in order to achieve high carrier densities without risking damaging our device, we make use of the combination of top and bottom gates, which lead to an induced charge carrier density equal to $\Delta n = \epsilon_{hBN} \epsilon_0 (V_{top} + V_{back}) / (e d)$. The largest possible voltage we can apply, limited by the dielectric breakdown of hBN of 0.7V/nm [see Ranjan et. al, *ACS Appl. Elect. Mater.* **3**, 3547 (2021)], is 14 V, which results in a doping level of 2.9x10¹³/cm². Our maximum doping level achieved, 1.8x10¹³cm², is well within this limit, and therefore we find it a reasonable value.

3. *Given the large tilting angle of the sample, which gives rise to external field anisotropy, I think it will make more sense to plot Fig. 3c&d with the effective magnetic field (or other fields that take into account of the sample tilting angle), instead of the external field. It's otherwise unclear what is the impacts of the sample tilting for this field anisotropy discussion.*

In our discussion we have decided to focus on the behavior of the magnetization precession amplitude and initial phase with respect to the external magnetic field. We have taken this decision since the magnetic anisotropy and magnetization tilting angle can vary throughout the magnetization dynamics. Nonetheless, following the reviewer's suggestion, we have tried replotting Fig. 3c and d *versus* H_{eff} (see Fig. R2 below). In our opinion, it does not lead to an easier interpretation. Plotting our data as a function of the in-plane and out-of-plane components of H_{eff} or H_{ext} would also not facilitate interpretation. Therefore, we prefer to keep Fig. 3c and d in their original form. We hope that the reviewer agrees with our decision.

Fig. R2: Magnetization precession amplitude and initial phase as a function of the effective magnetic field.

4. *To have sizable ICME effect, the field strength of pulsed laser must be large. Given the laser fluence in this experiment, do the authors estimate a large enough ICME to give rise to any observable effects? ICME should also have a specific fluence dependence. In the supplementary figure, the fluence dependence in Fig. S5 seems to indicate a relatively small effect on the magnon amplitude and phase.*

Indeed, the fluence used in literature for observing the ICME in garnet films is at least about a factor of 100 higher than the one we used in our studies – see Ref. 71, which uses 3 mJ/cm². Nonetheless, we would like to point out that the magnetic van der Waals semiconductor Cr₂Ge₂Te₆ is very different from traditional garnets in their light matter interaction, which allows us to observe the ICME even at low pump fluences. One key element is that the coupling between the magnetization and the electronic structure is very strong in CGT, producing a Faraday rotation of about 100 degrees/μm [*Phys. Rev. B* **98**, 1254196 (2018)]. This is multiple orders of magnitude larger than the 0.0175 degrees/μm for undoped yttrium iron garnet (YIG) [e.g. *J. Eur. Ceram. Soc.*

40, 6073-6078 (2020)]. This supports our observation of ICME in our samples using the pump fluence that we used.

The effective magnetic field generated by the ICME is linear in the intensity of the light, so the rotation of the magnetization due to an ultrashort ICME pulse is linear with the fluence of the laser pulse. The same holds true for the change in magnetization due to thermal demagnetization (ΔK mechanism) and photo-induced magnetic anisotropy for low fluences. For larger fluences, the change in magnetization becomes nonlinear due to e.g. the saturation of the absorption of the laser, and photodoping that results in a change in the opto-magnetic coefficients of the CGT. Moreover, the TRFE signal is also affected by the change in magnitude of the magnetization due to heating by the pump pulse.

To address this point of the reviewer and illustrate the effect of the pump fluence on the amplitude, we have extended Fig. S6 of the revised Supplementary Information to include the amplitude and starting phase of the magnetization precession (see Fig. R3 below). Supplementary Fig. 6b shows that the amplitude initially increases linearly, but saturates around a fluence of about $35 \mu\text{J}/\text{cm}^2$. Note that all data sets go through the origin, so the fluence does have a large effect on the amplitude.

The phase displayed in Supplementary Fig. 6c is practically independent of the pump fluence, which indicates that the ratio of the strengths of the dominant effects that are involved in the excitation of the magnetization precession does not depend much on the fluence.

From the fluence dependence of the TRFE amplitude and phase alone, we cannot conclude that non-thermal effects such as the ICME play a role in the excitation of the magnetization dynamics. However, in our manuscript we show several pieces of supporting evidence, see also our detailed response to reviewer 2.

We have added the discussion of the pump fluence dependence of the TRFE amplitude in Section 6 of the Supplementary Information.

Changes to the manuscript: In the Supplementary Information, we have added a discussion on the pump fluence dependence of the TRFE amplitude according to the discussion above. We have also updated Fig. 6 of the Supplementary Material according to Fig. R3 below.

Fig. R3: Frequency, amplitude, and initial phase of the magnetization precession as a function of the pump laser fluence.

Reviewers' Comments:

Reviewer #1:

Remarks to the Author:

The revised manuscript and response letter have answered all the questions/comments raised by this reviewer. I recommend it for publication.

Reviewer #2:

Remarks to the Author:

My questions have been properly addressed in the rebuttal letter and the revised manuscript. I recommend publication in Nature Communications.

Reviewer #3:

Remarks to the Author:

The authors addressed most of my comments, except for the comment on doping level estimate. As explained in the response letter, the authors were using a higher hBN dielectric in the estimate, which gives rise to 40% higher doping estimate. While there can be different reports on the values of hBN dielectric, the estimate needs to match results that can provide more accurate doping values, e.g. from Landau level measurements and Moire state filling. The authors should refer to those manuscripts and use an appropriate doping estimate. Once this issue is addressed, I can recommend its publication.

Reviewer #1

The revised manuscript and response letter have answered all the questions/comments raised by this reviewer. I recommend it for publication.

Reviewer #2

My questions have been properly addressed in the rebuttal letter and the revised manuscript. I recommend publication in Nature Communications.

Reviewer #3

The authors addressed most of my comments, except for the comment on doping level estimate. As explained in the response letter, the authors were using a higher hBN dielectric in the estimate, which gives rise to 40% higher doping estimate. While there can be different reports on the values of hBN dielectric, the estimate needs to match results that can provide more accurate doping values, e.g. from Landau level measurements and Moire state filling. The authors should refer to those manuscripts and use an appropriate doping estimate. Once this issue is addressed, I can recommend its publication.

We appreciate the comment by the reviewer. We note that the exact value of the gate capacitance is quite sensitive to the quantum capacitance contributions of the semiconductor channel which would then lead to a change on the effective dielectric constant estimated using the parallel plate capacitor model. Unfortunately, there are no measurements of the capacitance for our system in particular. Therefore, to address this point, we modified the values in the manuscript using a hBN dielectric constant of 3, in agreement with quantum magneto-transport using hBN-encapsulated graphene [Phys. Rev. Lett. 126, 156802 (2021)] and direct capacitance measurements in exfoliated hBN flakes with various thicknesses [Mater. Res. Express 9 065901 (2022)]. The change of ~20% in the values of dielectric constant compared to the one we used previously does not change any of the main results or the conclusions of our work and only leads to a similar change in the calculated values for the doping and electric field.

Changes in the manuscript:

We have modified all the values quoted for the calculated induced charge carrier density and electric field, including the main text, supplementary information and figures.

At the end of the 'Methods: Calculating Δn and ΔD from the gate voltages' section, we have added two short sentences in which we compare our maximum obtainable Δn with values reported in other works to show that our values are reasonable. The sentences start with "Filling in the ... as a dielectric", at the top of page 10.

We have included the above references when quoting the dielectric constant of hBN [Phys. Rev. Lett. **126**, 156802 (2021) and Mater. Res. Express **9**, 065901 (2022)].

We have gone through the manuscript once again and made small modifications to comply with the editorial requests - e.g. section titles and typos.